# Association between EEG Paroxysmal Abnormalities and Levels of Plasma Amino Acids and Urinary Organic Acids in Children with Autism Spectrum Disorder

**DOI:** 10.3390/children9040540

**Published:** 2022-04-11

**Authors:** Daniele Marcotulli, Chiara Davico, Alessandra Somà, Guido Teghille, Giorgio Ravaglia, Federico Amianto, Federica Ricci, Maria Paola Puccinelli, Marco Spada, Benedetto Vitiello

**Affiliations:** 1Section of Child and Adolescent Neuropsychiatry, Department of Public Health and Pediatric Sciences, University of Turin, 10126 Turin, Italy; daniele.marcotulli@unito.it (D.M.); alessandra.soma@unito.it (A.S.); guido.teghille@unito.it (G.T.); giorgio.ravaglia@edu.unito.it (G.R.); federica.ricci@unito.it (F.R.); benedetto.vitiello@unito.it (B.V.); 2Department of Neuroscience, University of Turin, 10126 Turin, Italy; federico.amianto@unito.it; 3Department of Clinical Biochemistry, “Baldi e Riberi” Laboratory, AOU Città della Salute e della Scienza di Torino, 10126 Turin, Italy; mpuccinelli@cittadellasalute.to.it; 4Section of Pediatrics, Department of Public Health and Pediatric Sciences, University of Torino, 10126 Turin, Italy; marco.spada@unito.it

**Keywords:** autism, ASD, EEG, metabolomic, amino acids, urinary organic acids

## Abstract

Abnormalities in the plasma amino acid and/or urinary organic acid profile have been reported in autism spectrum disorder (ASD). An imbalance between excitatory and inhibitory neuronal activity has been proposed as a mechanism to explain dysfunctional brain networks in ASD, as also suggested by the increased risk of epilepsy in this disorder. This study explored the possible association between presence of EEG paroxysmal abnormalities and the metabolic profile of plasma amino acids and urinary organic acids in children with ASD. In a sample of 55 children with ASD (81.8% male, mean age 53.67 months), EEGs were recorded, and 24 plasma amino acids and 56 urinary organic acids analyzed. EEG epileptiform discharges were found in 36 (65%) children. A LASSO regression, adjusted by age and sex, was applied to evaluate the association of plasma amino acids and urinary organic acids profiles with the presence of EEG epileptiform discharges. Plasma levels of threonine (THR) (coefficient = −0.02, *p* = 0.04) and urinary concentration of 3-Hydroxy-3-Methylglutaric acid (HMGA) (coefficient = 0.04, *p* = 0.02) were found to be associated with the presence of epileptiform discharges. These results suggest that altered redox mechanisms might be linked to epileptiform brain activity in ASD.

## 1. Introduction

Autism Spectrum Disorder (ASD) is a heterogeneous neurodevelopmental condition characterized by early onset deficit in social communication and reciprocity, together with a restricted range of interests and repetitive patterns of behavior [1]. Its pathogenesis is not fully understood, but complex interactions between multiple genes, epigenetic factors and exposure to environmental modifiers likely influence the expression of the disorder [2]. No diagnostic biomarker is currently available, and the diagnosis of ASD remains at this time entirely clinical.

Given the considerable genetic and phenotypic heterogeneity of ASD, efforts have been ongoing to identify more pathogenetically homogeneous subgroups that could be more easily targeted with specific biological interventions. Thus, “syndromic ASD” has been defined by associations with dysmorphic characteristics or other signs or symptoms such as epilepsy, intellectual disability, or motor deficits [3]. A growing body of literature shows that a certain number of ASD cases are associated with identifiable metabolic abnormalities, with potential implications for prevention or treatment [4]. In fact, some inborn error of metabolism (IEM) could be prevented, especially when there are high levels of consanguinity, and in some cases early treatment can result in better outcome [5,6].

IEMs are studied as a possible cause of ASD, and non-specific abnormalities of plasma amino acids and/or urinary organic acids have been investigated with the purpose of identifying potential biomarkers and clues to the pathogenesis of ASD [7]. Metabolomics, which is the systematic identification and quantitation of all metabolites in a given organism or biological sample [8], has been applied to studying ASD. Metabolomics studies in cohort of patients with ASD have yielded generally inconsistent and sometimes contradictory results, due to the different methods used to perform metabolomic analysis and the small size and heterogeneity of the study samples, which varied with respect to age, sex, nd potential pharmacological confounders [9,10]. Glutamate, GABA, and/or glutamine have been among the most studied amino acids for their role in neurotransmission [11,12,13,14,15,16,17].

Many studies have tried to identify metabolic features in plasma that can help differentiate between ASD and typical development. To this end, univariate, multivariate, and machine learning methods have been used [18,19]. Research has focused on identifying combinations of several metabolites rather than single metabolites and on the accuracy with which these combinations can discriminate between normal and ASD populations. Among others, Orozco and Colleagues analyzed the plasma metabolic profile of ASD, idiopathic-developmental delay, and Down syndrome in children as compared with typically developing controls, and found a perturbation in one-carbon metabolism pathways (i.e., the metabolic pathways converging at homocysteine metabolism, glutathione biosynthesis, folate cycle, and choline/betaine metabolism) [19]. However, a small number size is a main limitation of these studies [18,20]. Smiths et al. analyzed a large population from Children’s Autism Metabolome Project and stratified ASD subjects into subgroups based on shared metabolic phenotypes associated with branched chain amino acids (BCAA) dysregulation [21]. The combination of glutamine, glycine, and ornithine identified a dysregulation in amino acids metabolism that was present in 16.7% of the ASD subjects with a specificity of 96.3% and a PPV of 93.5% [21]. In a subsequent analysis from the same project, thirty-four candidate metabotypes were identified that differentiated subsets of ASD from typically developing participants, forming six metabolic clusters based on ratios of either lactate or pyruvate, succinate, glycine, ornithine, 4-hydroxyproline, or α-ketoglutarate with other metabolites [22].

A number of hypotheses have been proposed to explain the possible pathogenetic effect of metabolic abnormalities in ASD. Most hypotheses involve complex interactions between inflammatory response, gut-brain axis, and altered redox state [23]. Indeed, accumulating evidence suggests that altered bidirectional interactions between the central nervous system and the gastrointestinal tract (brain-gut axis) may have a role in ASD [24], with possible implications for the development of microbiome-mediated therapeutic interventions [25].

Among the hypothesized pathogenetic mechanisms in ASD, altered redox appears to be especially relevant since increased oxidative stress in the brain may have functional consequences in terms of a chronic inflammatory response, increased mitochondrial superoxide production, and oxidative protein and DNA damage [26,27]. The redox status can be defined as the balance between cellular oxidants species (i.e., free radicals and other reactive species) and antioxidant capability: a disruption of this equilibrium toward an oxidized state results in oxidative stress, with effect on immune system activation, regulation of mitochondrial function, cell cycle regulation, cell differentiation, enzyme activity regulation and many other consequences. An imbalance between excitation and inhibition has also been suggested as a mechanism to explain dysfunctional brain networks in ASD [28,29,30,31], and might be linked to metabolic abnormalities. An inhibition/excitation imbalance is also consistent with the increased risk of epilepsy [32,33,34] and the frequent finding of EEG paroxysmal abnormalities in ASD [35,36,37].

This study was aimed at examining the relationship between the presence of EEG paroxysmal abnormalities and the metabolic profile of plasma amino acids and urinary organic acids in a sample of children with ASD.

## 2. Materials and Methods

### 2.1. Design and Participants

This was a cross-sectional study of 55 consecutively referred children (mean age 53.67 ± 37.42 months, range 25–203 months, corresponding to 2–16 years) at the University of Turin-Pediatric Hospital Regina Margherita Outpatient Service for Neurodevelopmental Disorders. Following a comprehensive evaluation conducted by trained child psychiatrists, the children met DSM-5 diagnostic criteria for ASD. Parents gave informed permission to participate, and all the procedures were approved by the local ethics committee. All the children received the following assessments.

### 2.2. Plasma Amino Acids

EDTA plasma from fasting patients was collected and deproteinized with sulfosalicylic acid. Specimens were stored at −20 °C until analyzed. The amino acids profile was obtained with Biochrom 20Plus Amino Acid Analyser based on ion exchange chromatography with post column derivatization with Ninhydrin [38,39]. Ninhydrin derivatives were analyzed at 2 wavelengths (i.e., 570 and 440 nm) to assess peak purity. Quantitation was performed with two internal standards (norleucine and amino-ethyl-cystine) and an external calibration curve (Amino Acid Standards Physiological^®^ by Sigma Aldrich, St. Louis, MO, USA). In each analytical series were analyzed as Internal Quality Control, AMI.02-1 and AMI.02-2 (Control Amino Acid MCA—The Netherlands). The laboratory is certified ISO 9001:2015 and participates in Erndim international External Quality Schemes. Eighty metabolic variables (24 plasma amino acids and 56 urinary organic acids) were collected.

### 2.3. Urinary Organic Acids

Urines from the first morning void were collected: creatinine was determined in fresh urine and then samples were stored at −20 °C until analysis. Organic acids and an acylglycine profile were obtained by Gas Chromatography coupled with Mass Spectrometry [38,39]. Urine was diluted to obtain a concentration of 2 mM of creatinine and the oximation of α-ketoacids was performed. Extraction of organic acids and acylglycines was carried out with ethyl acetate from the acidified sample. The dried extracts were added with a derivatization agent to obtain trimethylsilyl of metabolites. The analysis was accomplished on a GC-MS equipped with an Electron Ionization (EI) source. MS spectra were acquired by Scan Mode and a simultaneous Selected ion monitoring (SIM) mode was used to increase sensitivity of less abundant clinical important molecular species (e.g., 3-hydroxyglutaric, succinylacetone, mevalonolactone, most acylglycines). Two internal standards were used: 2-keto-caproic for 2-keto acids, and tropic acids for other acids and acylglycines.

Quantitative analyses were performed with external calibration using a home-made standard curve in the urine matrix. Samples concentrations were normalized to urine creatinine content. In each analytical series, ORG.02-1 and ORG02.2 (Control Organic Acid MCA—The Netherlands) were analyzed as internal quality control. The laboratory is certified ISO 9001:2015 and participates in Erndim international External Quality Schemes.

### 2.4. Electroencephalography

EEGs were obtained with the international 10–20 system of electrode placement. The recording included hyperventilation and photic stimulation. Trained pediatric neurologists evaluated the EEGs for presence and localization of paroxysmal abnormalities (epileptiform discharges).

### 2.5. Statistical Analysis

Statistical analyses were performed using the statistical programming language R (version 4.0.5) [40]. Descriptive statistics was applied to the sociodemographic and clinical data. Continuous variables were described by mean and SD, and categorical data as percentages. A z-test was used to evaluate differences between proportions of categorical variables. Group differences for continuous variables were assessed with a two-tailed Mann–Whitney U test. A *p*-value <0.05 was considered statistically significant.

Given the high number of metabolic variables that were tested (24 plasma amino acids and 56 urinary organic acids), we employed the least absolute shrinkage and selection operator (LASSO) penalized regression for selecting the variables predictive of abnormal EEG. Children’s age and sex were also considered in the same model as adjunctive predictors. λ value parameter values for L1-penalized least absolute shrinkage were selected using 5-fold cross-validation based on AUC in *glmnet*. The method proposed by Liu and colleagues was used to calculate confidence intervals and make inferences [41].

## 3. Results

A total of 55 children, 81.8% males, participated in the study. Their characteristics are presented in Table 1: the mean age was 53.67 months (SD 37.42, range 25–203 months). Five children had a diagnosis of epilepsy, and one a diagnosis of febrile seizure plus (FS+). Two of the patients were receiving pharmacological treatment at the time of the study (both of them were on valproic acid, and one was also on clobazam).

EEG paroxysmal abnormalities were found in 36 children (65.45%). This group with EEG paroxysmal abnormalities was younger (47.39 ± 36.8 months) than the group with normal EEG (65.58 ± 37.96 months) (*p* = 0.04), with no sex difference (Table 1). The most common localization of paroxysmal abnormalities was fronto-central (*n* = 18, 50%) (Table 2). Other localizations were multifocal/diffuse (*n* = 4, 11.11%), central (*n* = 3, 8.3%), frontal (*n* = 3, 8.3%), fronto-temporal (*n* = 3, 8.3%), centro-temporal (*n* = 3, 8.3%), and temporo-occipital (*n* = 2, 5.5%) (Table 3).

We then evaluated whether the combined plasma amino acids and urinary organic acids profiles differed between patients with and without paroxysmal EEG abnormalities using LASSO regression with post-selection inference. The model was also adjusted by the patient’s sex and age. Fifteen important variables (i.e., variables with coefficients not shrunk to zero) were identified (Table 4). A distance from zero of the coefficients informs the magnitude of the importance in the prediction model. Of the fifteen variables identified, THR and HMGA were found to be significantly associated with the presence of EEG epileptiform discharges. In particular, lower circulating levels of THR (coefficient −0.02, *p* = 0.04) and higher urinary concentrations of HMGA (coefficient = 0.04, *p* = 0.02) were associated with epileptiform discharges on the EEG (Table 4).

## 4. Discussion

This exploratory study found that the presence of abnormalities in brain activity as shown by EEG epileptiform discharges was associated with lower plasmatic levels of THR, and higher urinary levels of HMGA in children with ASD. Lower levels of THR in ASD have been documented compared with normal controls [42,43,44]. Previous research has suggested that THR plasma levels can affect the neurotransmitter balance in the brain [45,46]. Moreover, a deficiency in THR has been associated with increased seizure susceptibility in mice [47,48], and a diet supplemented with THR and other ketogenic amino acids has been shown to reduce seizure susceptibility [49]. Thus, the data from our study are consistent with a number of previous studies in both humans and rodents, and provide further support to the possible pathogenetic role of a deficiency of THR in the brain hyperexcitability seen in ASD. Lower THR levels were also found in another group of children with ASD and EEG epileptic abnormalities, compared with ASD without those abnormalities, together with significantly lower plasma levels of glycine, histidine, ornithine, lysine, α-aminobutyric acid, and arginine, and higher plasma levels of asparagine [50].

The elevated urinary concentration of HMGA found in our study in the ASD patients with paroxysmal abnormalities is also consistent with previous studies [51,52], and suggests that altered brain activity may be associated with mitochondrial dysfunction.

Numerous studies have highlighted the importance of the epigenetic mechanisms and the role of DNA methylation in affecting ASD phenotypes [52,53]. Bam et al. found HGMA to be among the urinary metabolites that most correlate with the mitochondrial DNA (mtDNA) copy number. An increase in mtDNA copy number is known to be a compensatory effect to mitochondrial dysfunction. Moreover, HMGA has been found to impair mitochondrial function in the rat brain by decreasing activities of glutathione peroxidase and citric acid cycle enzymes. This organic acid induces oxidative stress and disrupts mitochondria bioenergetics, dynamics and ER-mitochondria crosstalk [54]. Oxidative stress has been reported in patients with deficiency of the mitochondrial matrix enzyme that catalyzes the last step of ketogenesis and leucine catabolism (3-hydroxy-3-methylglutaryl coenzyme A lyase), leading to elevated HMGA urinary levels [55].

A deeper understanding of the role of mitochondrial dysfunction in the pathogenesis of ASD may lead to new therapeutic targets. Delhey and colleagues have reported clinical improvement in children with ASD by boosting the mitochondrial activity of complex I, complex IV and citrate synthase with the administration of fatty acid, folate and B12 [56]. A ketogenic diet, which is known to improve autistic behaviors in both humans and rodents, was shown to recover both mitochondrial function and morphology in mice [57]. Finding a depletion of THR, which is known to have also a ketogenic effect, in children with epileptiform discharges further supports its possible pathogenetic relevance. The finding of an elevation of urinary HGMA in the present study can help understand the complex metabolic mechanisms underlying ASD as a step towards designing customized therapies for specific mitochondrial dysfunctions.

Although altered levels of both THR and HMGA have been associated with ASD and/or brain hyperexcitability, the link between the two molecules have not been studied yet and, to our knowledge, there is no metabolic pathway where both THR and HMGA interact. Indeed, it is likely that the altered levels of these molecules are part of wider metabolic abnormalities that stem from oxidative stress and altered amino acid metabolism and lead to brain hyperexcitability.

This study has several limitations. First, the study evaluated a relatively small sample of ASD children, without including a comparison group of typically developing children. Second, the study sample had a male preponderance, which is consistent with the greater prevalence of ASD in the male population, but still limits inferences to female patients. Third, the EEG clinical evaluation was not accompanied by quantitative EEG measures, and thus the identification of paroxysmal abnormalities was subject to human error. Moreover, no standardized measures of the intensity and frequency of paroxysmal abnormalities were available. Fourth, although regularization techniques can help in reducing the number of variables, drawing post-selection inference on a large number of variables with a lower number of subjects is still challenging, and the results should be carefully considered in light of this limitation.

Further studies with larger samples and more in-depth EEG methodology are needed to confirm and better clarify our findings.

## 5. Conclusions

In conclusion, in an exploratory study of the relationship between presence of EEG epileptiform abnormalities and plasma and urine metabolic profiles in children with ASD, statistically significant associations were found with lower plasma levels of the amino acid THR and higher urinary levels of HGMA. These data, taken together with previous reports in humans and rodents, further support these metabolic characteristics as possible mediators of an excitatory/inhibitory imbalance in ASD, and can contribute to identifying more biologically homogeneous subgroups within the ASD clinical construct.

## Figures and Tables

**Table 1 children-09-00540-t001:** Demographics and clinical characteristics.

	All *n* = 55	with Abnormal EEG *n* = 36	with Normal EEG *n* = 19	*p*-Value
Males, *n* (%)	45 (81.8)	30 (83.3)	15 (78.9)	0.97
Age at evaluation, months, mean (SD) years	53.67 (37.42)	47.39 (36.08)	65.58 (37.96)	**0.04**
	[M1] 4.5 (3.1)			
Epilepsy or FS+, *n* (%)	6 (11.1)	6 (16.7)	0 (0)	

Statistically significant values are bold.

**Table 2 children-09-00540-t002:** EEG findings.

Localization	Patients with EEG Abnormalities, *n* = 36 *n* (%)
Multifocal or diffuse	4 (11.1)
Fronto-central	18 (50.0)
Frontal	3 (8.3)
Central	3 (8.3)
Centro-temporal	3 (8.3)
Fronto-temporal	3 (8.3)
Temporo-occipital	2 (5.6)

**Table 3 children-09-00540-t003:** Assessed plasma amino acids and urinary organic acids.

Plasma Amino Acids	Urinary Organic Acids	
s-Aspartic acid s-Hydroxyproline s-Threonine s-Serine s-Asparagine s-Glutamic acid s-Glutamine s-Proline s-Glycine s-Alanine s-Citrulline s-α-Aminobutyric acid s-Valine s-Cystine s-Methionine s-Isoleucine s-Leucine s-Tyrosine s-Phenylalanine s-Ornithine s-Lysine s-Histidine s-Tryptophan s-Arginine	u-Lactic acid u-Glycolic acid u-Glyoxylic acid u-2-Hydroxybutyric acid u-Oxalic acid u-Pyruvic acid u 3-Hydroxybutyric acid u 3-Hydroxyisobutyric acid u Malonic acid u 3-Hydroxyisovaleric acid u Methylmalonic acid u2-Ethyl-3-Hydroxypropionic acid u 4-Hydroxybutyric acid u Ethylmalonic acid u Phenylacetic acid u Succinic acid u Methylsuccinic acid u Glyceric acid u Fumaric acid u 5-Hydroxyhexanoic acid u Isobutyryl Glycine u Propionyl Glycine u Mevalonolactone u Glutaric acid u 3-Methylglutaric acid u 3-Methylglutaconic acid u Glutaconic acid u Mandelic acid	u Isovalerylglycine u Pyroglutamic acid u 7-Hydroxyoctanoic acid u Tiglil Glycine u 2-Hydroxyphenylacetic acid u 2-Hydroxyglutaric acid u 3-Hydroxyglutaric acid u Phenyllactic acid u3-Hydroxy-3-Methylglutaric acid u 4-Cyclohexylacetic acid u 2-ketoglutaric acid u 4-Hydroxyphenylacetic acid u Hexanoyl Glycine u N-Acetylaspartic acid u 2-Hydroxyadipic acid u Octenedioic acid u 3-Hydroxyadipic acid u Suberic acid u 2-Ketoadipic acid u cis_Aconitic acid u Citric acid u Methylcitric acid u 4-Hydroxyphenyllactic acid u4-Hydroxyphenylpyruvic acid u N-Acetyl Tyrosine u Hexadecanedioic acid u Malic acid u Adipic acid

**Table 4 children-09-00540-t004:** Least absolute shrinkage and selection operator (LASSO) penalized regression analyses: important variables for EEG-based classification.

	Coefficients	IC (95%)		*p*-Value
Threonine THR (p)	−0.02	**−0.08**	**−0.01**	**0.04**
Serine (p)	−0.00	−0.01	0.08	0.17
Asparagine (p)	−0.00	−0.01	0.01	0.67
Glutamine (p)	−0.00	−0.01	0.01	0.59
Alanine (p)	0.00	−0.00	0.02	0.28
Citrulline (p)	0.00	−0.07	0.09	0.83
Valine (p)	0.00	−0.02	0.02	0.84
Leucine (p)	0.00	0.01	0.12	0.06
3-Hydroxyisobutyric acid (u)	0.00	−0.05	0.12	0.52
3-Hydroxyisovaleric acid (u)	0.01	−0.02	0.09	0.28
Succinic acid (u)	0.01	−0.02	0.07	0.30
Pyroglutamic acid (u)	0.01	−0.01	0.04	0.22
3-Hydroxy-3-methylglutaric acid HMGA (u)	0.04	**0.02**	**0.13**	**0.02**
2-Ketoglutaric acid (u)	−0.01	−0.03	0.00	0.24
Citric acid (u)	−0.00	−0.00	0.00	0.69

Statistically significant values are bold.

## Data Availability

Derived data supporting the findings of this study will be available from the corresponding author upon reasonable request. The raw data are not publicly available due to privacy considerations.

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
