# Peer review of "Association between EEG Paroxysmal Abnormalities and Levels of Plasma Amino Acids and Urinary Organic Acids in Children with Autism Spectrum Disorder"

_children, 2022, doi:10.3390/children9040540_

Round 1

Reviewer 1 Report

This paper seeks to find correlations between metabolites and abnormal EEG readings in children with ASD. Overall, this is an interesting topic and I agree with the finding that there are likely signatures in metabolite profiles that can be found that  are tied to certain co-occurring conditions of ASD. That being said, these studies generally face the challenge that the number of measurements per participant are quite large compared to the number of clinical trial participants and this study is no exception to this.

Overall, the paper is well structured and the results are clearly presented. That being said, some specific comments are given below:

  • While the paper contains a good number of reference, I am somewhat surprised that there are relatively few reference related to differences in metabolite patterns in children with ASD. Given that this topic is a core theme of this work, there should be a paragraph in the introduction that discusses the literature in this field in more depth. I would especially recommend to include a discussion of multivariate analysis of metabolites as these metabolites are linked to each other in biological pathways (i.e., it is unlikely that differences in individual metabolites are predictive, but combinations of several metabolites might be able to do so).
  • One of the most significant challenges related to these types of investigation is that one uses models that discriminate different outcomes (abnormal EEG vs normal EEG) based upon a significant number of inputs. Ideally, one would like to have 10-20 data points (i.e., clinical trial participants) per parameter to be estimated (where each input results in at least one parameter). However, these studies often have as many or more potential inputs as they have clinical trial participants. This is also the case in this work where 55 children are investigated and 80 measurements are used per child (24 plasma amino acids and 56 urinary organic acids). Regularization techniques can help, and also do so in this work, but there are limits to what can be achieved. If I understood this correctly, then 15 inputs are still identified as part of this study. That number is quite a bit larger than I would deem acceptable, given that the study includes 55 children. As a minimum the work should include a more detailed analysis of how this changes based upon how regularization parameters are chosen.
  • The work should include a more detailed discussion of how the identified metabolites are correlated/linked to each other in biological pathways (i.e., the current discussion is a little less than 1 page in length only).
  • There are some inconsistencies in the reference list that should be corrected. Some references use a different format than others and the last reference is missing the reference number.

Author Response

Reviewer 1

This paper seeks to find correlations between metabolites and abnormal EEG readings in children with ASD. Overall, this is an interesting topic and I agree with the finding that there are likely signatures in metabolite profiles that can be found that  are tied to certain co-occurring conditions of ASD. That being said, these studies generally face the challenge that the number of measurements per participant are quite large compared to the number of clinical trial participants and this study is no exception to this.

Overall, the paper is well structured and the results are clearly presented. That being said, some specific comments are given below:

  • While the paper contains a good number of references, I am somewhat surprised that there are relatively few references related to differences in metabolite patterns in children with ASD. Given that this topic is a core theme of this work, there should be a paragraph in the introduction that discusses the literature in this field in more depth. I would especially recommend to include a discussion of multivariate analysis of metabolites as these metabolites are linked to each other in biological pathways (i.e., it is unlikely that differences in individual metabolites are predictive, but combinations of several metabolites might be able to do so).

Thank you for your suggestion, we expanded this point in the introduction section. 

The following paragraph was added (page 2): 

“Many studies tried to identify metabolic features in plasma that can help differentiate between ASD and typical development people. In order to achieve this goal, univariate, multivariate and machine learning methods have been used (West, 2014; Orozco, 2019). Research has focused on identifying combinations of several metabolites rather than single metabolites and describing accuracy with which these combinations are able to distinguish typical from ASD population. Among others, Orozco and Colleagues analyzed plasma metabolic phenotype of ASD, idiopathic-developmental delay and Down syndrome children as compared to typically developed controls and found perturbation in one-carbon metabolism pathways (i.e. metabolic pathways converging at homocysteine metabolism, glutathione biosynthesis, folate cycle, and choline/betaine metabolism) (Orozco, 2019). Being small number of the samples one of the main limitations of these researches  (West, 2014; Rangel-Huerta, 2019), Smiths and Colleagues analyzed a large population from Children’s Autism Metabolome Project, in order to stratify ASD subjects into subpopulations based on shared metabolic phenotypes associated with branched chain amino acids (BCAA) dysregulation. The combination of glutamine, glycine, and ornithine identified a dysregulation in amino acids metabolism that was present in 16.7% of the ASD subjects and is detectable with a specificity of 96.3% and a PPV of 93.5% (Smiths, 2019). In a subsequent work from the same project, Authors identified thirty-four candidate metabotypes that differentiated subsets of ASD from typically developing participants, forming six metabolic clusters based on ratios of either lactate or pyruvate, succinate, glycine, ornithine, 4-hydroxyproline, or α-ketoglutarate with other metabolites (Smiths, 2020).”

  • One of the most significant challenges related to these types of investigation is that one uses models that discriminate different outcomes (abnormal EEG vs normal EEG) based upon a significant number of inputs. Ideally, one would like to have 10-20 data points (i.e., clinical trial participants) per parameter to be estimated (where each input results in at least one parameter). However, these studies often have as many or more potential inputs as they have clinical trial participants. This is also the case in this work where 55 children are investigated and 80 measurements are used per child (24 plasma amino acids and 56 urinary organic acids). Regularization techniques can help, and also do so in this work, but there are limits to what can be achieved. If I understood this correctly, then 15 inputs are still identified as part of this study. That number is quite a bit larger than I would deem acceptable, given that the study includes 55 children. As a minimum the work should include a more detailed analysis of how this changes based upon how regularization parameters are chosen.

We thank the reviewer for the insightful comment. We agree that dealing with a number of independent variables that exceeds the number of events/subjects is one of the most significant challenges in this work and in similar research. Accordingly, in the limitations we have stated that “Although regularization techniques can help in reducing the number of variables, drawing post-selection inference on a large number of variables with a lower number of subjects is still challenging and results should be carefully considered” (page 5).  We acknowledge that even consolidated approaches like regularization techniques need to be carefully used. For this reason, we chose the regularization parameter λ that yielded the greater AUC in five cross-validation splits, with alpha set to 1. The chosen λ  using this approach was 0.18. To evaluate the sensitivity of our results to the regularization parameter, we also checked whether the number of selected variables and the associated confidence intervals changed when using λ selected with different criterion (i.e. within one standard error from the minimum cross validation error) and found no significant differences in the selected predictors.

  • The work should include a more detailed discussion of how the identified metabolites are correlated/linked to each other in biological pathways (i.e., the current discussion is a little less than 1 page in length only).

Thank you for your suggestion, we included the following paragraph to enrich the discussion on this topic:  “While altered levels of both THR and HMGA have been associated with ASD and/or brain hyperexcitability, the link between the two molecules have not been studied yet and, to our knowledge, there is no metabolic pathway where both THR and HMGA interact. Indeed, it is likely that the altered levels of these molecules are part of wider metabolic abnormalities that stem from oxidative stress and altered amino acid metabolism and lead to brain hyperexcitability.”

  • There are some inconsistencies in the reference list that should be corrected. Some references use a different format than others and the last reference is missing the reference number.

Thank you for correcting the reference list

Author Response

Reviewer 2

  • Altered redox appears to be important and consider: elaborating on this term for the layperson. How does it connect?

Thank you, we explained this issue more in detail. The following paragraph was included in the manuscript “The redox status can be defined as the balance between cellular oxidants species (i.e., free radicals and other reactive species) and antioxidant capability: a disruption of this equilibrium toward an oxidized state results in oxidative stress, with effect on immune system activation, regulation of mitochondrial function, cell cycle regulation, cell differentiation, enzyme activity regulation and many other consequences.” (page 2).

  • though it makes sense to keep everything as months, it is easier for the reader to say 2-16 years instead of 25-203 months.

Yes, thank you for the suggestion. We have added the corresponding age in years in the text.

  • Is there any differences between studies you refer to and if so can you contrast it with yours? This may strengthen your article. 

Thank you. The following paragraph was added to the discussion section: “ Lower THR levels were also found in  another group of children with ASD and EEG epileptic abnormalities, compared with ASD without those abnormalities, together with significantly lower plasma levels of glycine, histidine, ornithine, lysine, α-aminobutyric acid, and arginine, and higher plasma levels of asparagine (Saleem, 2020).” 

  • This limitation can be made specific to include "even though females with ASD are not commonly seen in clinical practice, this finding may help in screening in females suspected of ASD". Consider stating other gender neutral population were not included.

Thank you for your comment. While markers that help in screening females suspected of ASD might be useful, we believe that the low number of females in our sample cannot support conclusions on females. As for other gender neutral population, we considered biological sex rather than gender, because of possible implications on metabolites levels. For clarifying this point, “(sex)” has been specified when reporting the percentage of male in our sample in the results section and table1.

  • As I came across this limitation I looked back to see the method. It may be best if you can be specific about EEG limitations. Is this clinically the most common EEG done for this population or did you do more. Real life limitations keep the article authentic and genuine which in turn helps the reader. In other words, contrast it with clinical work.

Thank you for this suggestion. In the methods section, under the EEG heading, we have specified that EEGs were performed in the context of routine clinical practice.  In the limitation paragraph, we have revised the text according to your suggestion by stating that “EEG clinical evaluation was not accompanied by quantitative EEG measures, thus the identification of paroxysmal abnormalities was subject to human error. Moreover, no standardized measures of the intensity and frequency of paroxysmal abnormalities were available.”

Round 2

Reviewer 1 Report

The authors have appropriately addressed my concerns.